# Frontiers of Collaboration between Primary Care and Specialists in the Management of Metabolic Dysfunction-Associated Steatotic Liver Disease: A Review

**DOI:** 10.3390/life13112144

**Published:** 2023-10-31

**Authors:** Koki Nagai, Kazuki Nagai, Michihiro Iwaki, Takashi Kobayashi, Asako Nogami, Masanao Oka, Satoru Saito, Masato Yoneda

**Affiliations:** 1Gastroenterology Division, National Hospital Organization Yokohama Medical Center, 3-60-2 Harajyuku, Totsuka-ku, Yokohama 245-8575, Japan; k78naga@gmail.com; 2Nagai Clinic, 1-7-25 Yokodai, Isogo-ku, Yokohama 235-0045, Japan; k-nagai@mtj.biglobe.ne.jp; 3Department of Gastroenterology and Hepatology, Yokohama City University Hospital, 3-9 Fuku-ura, Kanazawa-ku, Yokohama 236-0004, Japan; michihir@yokohama-cu.ac.jp (M.I.); tkbys@yokohama-cu.ac.jp (T.K.); nogamia@yokohama-cu.ac.jp (A.N.); 4OkaMedical, 1-19-18-3F Kamiookanishi, Kounan-ku, Yokohama 233-0002, Japan; okam@clinics.jp; 5Sanno Hospital, 8-10-16 Akasaka, Minato-ku, Tokyo 107-0052, Japan; ssai1423@iuhw.ac.jp

**Keywords:** nonalcoholic fatty liver disease, metabolic dysfunction-associated steatotic liver disease, review, management of MASLD, hepatologists, primary care physicians

## Abstract

Metabolic dysfunction-associated steatotic liver disease (MASLD), formerly known as nonalcoholic fatty liver disease (NAFLD), is the most common liver disease. It has a rapidly growing patient population owing to the increasing prevalence of obesity and type 2 diabetes. Patients with MASLD are primarily treated by family physicians when fibrosis is absent or mild and by gastroenterologists/hepatologists when fibrosis is more advanced. It is imperative that a system for the appropriate treatment and surveillance of hepatocellular carcinoma be established in order to ensure that highly fibrotic cases are not overlooked among the large number of MASLD patients. Family physicians should check for viral hepatitis, autoimmune hepatitis, alcoholic liver disease, and drug-induced liver disease, and should evaluate fibrosis using NIT; gastroenterologists/hepatologists should perform liver biopsy, ultrasound elastography (260 units in Japan as of October 2023), and MR elastography (35 units in Japan as of October 2023). This review presents the latest findings in MASLD and the role, accuracy, and clinical use of NIT. It also describes the collaboration between Japanese primary care and gastroenterologists/hepatologists in Japan in the treatment of liver diseases, including MASLD.

## 1. Introduction

Metabolic dysfunction-associated steatotic liver disease (MASLD) [1,2], formerly known as nonalcoholic fatty liver disease (NAFLD), and metabolic dysfunction-associated steatohepatitis (MASH) [1,2], the replacement term for nonalcoholic steatohepatitis (NASH), are recognized as the most common liver diseases. The number of patients with these diseases is rapidly increasing worldwide, owing to the increasing prevalence of obesity and type 2 diabetes [3,4,5]. Viral hepatitises, such as hepatitis B and C, have been controlled with the development of effective drugs [6], whereas cases of cirrhosis and hepatocellular carcinoma (HCC) caused by MASLD/MASH are increasing [7]. Therefore, MASLD/MASH is becoming increasingly recognized as a major burden to patients, healthcare providers, and the healthcare system as a whole and as an important medical/economic issue.

In MASLD, liver fibrosis is a major determinant of an adverse outcome and is associated with increased mortality in general and liver-related mortality in particular [8,9]. Patients with MASLD often become symptomatic with decompensated cirrhosis or are referred after HCC is detected. The selection of high-risk patients among the vast number of patients with MASLD is important from the perspective of life expectancy and health care economics and is considered an urgent issue [10]. However, from a medical economic point of view, it is difficult to perform invasive liver biopsy, frequent imaging tests, and specialized medical care in all MASLD cases. Furthermore, it is not possible for hepatologists to treat all patients with MASLD. In fact, most patients with fatty liver diseases, including MASLD, are followed up by primary care physicians and internists rather than hepatologists. Therefore, in recent years, several guidelines have been issued regarding how to deal with MASLD, not only for gastroenterologists/hepatologists, but also for diabetologists and primary care practice physicians [11,12,13]. These guidelines have made specialists and non-specialists widely aware of the disease concept and guidelines, including diagnosis and treatment. Most current guidelines recommend that patients with MASLD undergo risk stratification according to the presence or absence of significant fibrosis [11,12,13,14]. However, it is difficult to predict fibrosis progression and cirrhosis in routine medical care and physical examination. MASLD and MASH have few symptoms, and ALT levels in blood tests are not helpful. In addition, many patients with cirrhosis do not show a decrease in platelets or albumin and are often diagnosed in the advanced stages of cirrhosis, hepatocellular carcinoma, or ruptured esophageal varices. Therefore, screening methods are being investigated by which to efficiently and noninvasively identify cases of advanced liver fibrosis and reduce unnecessary referrals to hepatologists. It is also essential to establish a secondary check for the identification of high-risk patients with advanced fibrosis by the referring hepatologist, because high-risk patients should be followed up by hepatologists, and low-risk patients should be followed up by primary care physicians. Thus, the question of how hepatologists, primary care physicians, and internists view MASLD, and how they cooperate with each other, is an important one. This review presents the latest findings in MASLD and the role, accuracy, and clinical use of non-invasive testing (NIT). This review also introduces a strategy for the screening of advanced fibrosis cases in primary care in Japan, one that can contribute to the process of a thorough detailed examination by a gastroenterologist/hepatologist.

## 2. Concepts of NAFLD/NASH, MAFLD, and MASLD

Until the 1970s, the prevailing view was that fatty liver in the absence of alcohol consumption did not progress to cirrhosis [15]. In 1980, Ludwig et al. of the Mayo Clinic proposed a new disease concept, NASH, for a condition in which fatty liver inflammation occurs despite there being no history of alcohol consumption [16]. However, the concept of NASH did not receive substantial attention at that time [17,18]. Later, in 1985, Schaffner et al. proposed the concept of NAFLD as a general term for fatty liver disease not involving alcohol consumption [19]. Although NAFLD and NASH are described as “nonalcoholic” fatty liver diseases, the reality is that they affect non- to light drinkers with no obvious habitual drinking practices. Harmful drinking is defined as alcohol consumption of 30 g/day or more for men and 20 g/day or more for women [11,14]. NASH and NAFLD attracted attention again in 1998 when the National Institutes of Health in the United States declared the importance of NASH disease in regard to instances of cirrhosis of unknown cause and as obesity became a social problem.

NAFLD is classified into two major categories: nonalcoholic fatty liver (NAFL), in which the disease rarely progresses, and NASH, in which fibrosis tends to develop in the state of fatty hepatitis and can cause cirrhosis and HCC [11,14]. However, there is no clear distinction between NAFL and NASH, and there have been reports of cross-transference (Figure 1) [20]. NAFLD is based on the diagnosis of “fatty liver disease not caused by viral liver disease or alcohol,” which is a diagnosis of exclusion and does not reflect the actual etiology of the disease. In 2020, a consensus statement by 32 experts from 22 countries proposed MAFLD as a new disease concept to replace NAFLD [21]. MAFLD is diagnosed in patients with or without viral liver disease, with or without alcohol consumption, and with or without obesity, type 2 diabetes mellitus, or metabolic disorders. MAFLD is not an evidence-based diagnosis. However, considering the current pathology of fatty liver disease, “metabolic dysfunction” may be a more accurate description of the disease than “nonalcoholic” fatty liver disease. The validity of MAFLD needs to be confirmed in the future [21].

Recently, the disease concept of MASLD has also been proposed. At the European Association for the Study of the Liver (EASL) Congress 2023, La Asociación Latinoamericana para el Estudio del Hígado, American Association for the Study of Liver Diseases (AASLD), leaders of the multinational liver societies of the EASL, and co-chairs of the NAFLD Nomenclature Initiative proposed the term steatotic liver disease (SLD) as an overarching term to encompass the various etiologies of steatosis. With the implication of eliminating the discriminatory term “fatty,” NAFLD may be changed to MASLD and NASH may be changed to MASH in the future [1,2].

## 3. Pathogenesis of MASLD/MASH

The pathogenesis of MASLD/MASH is a complex interplay of environmental and genetic factors (Figure 2) [22]. The concept of multiple parallel hits rests on the hypothesis that many hits occur heterochronously in parallel, causing inflammation and fatty degeneration in the liver [22]. Dietary habits, insulin resistance, lipotoxicity, gut microbiota, and genetic factors play a central role in these parallel hits. Insulin resistance, in particular, is one of the most important factors in the pathogenesis of MASLD and MASH [23,24]. Insulin resistance leads to increased de novo lipogenesis in the liver and the suppression of lipolysis in adipose tissue, resulting in increased fatty acid production in the liver [25,26]. Insulin resistance also promotes adipose tissue dysfunction with altered production and secretion of adipokines and inflammatory cytokines [27]. MASLD is associated with decreased insulin sensitivity early in its course, even in the absence of overt diabetic complications [28]. Reportedly, 30% to 40% of patients with diabetes-associated MASLD have advanced MASH, are prone to liver fibrosis development, and frequently develop HCC [29].

Visceral adipose tissue is involved in MASLD development and in its exacerbation through direct effects, including the abnormal production of adipocytokines such as adiponectin, and indirect effects, such as increased insulin resistance [30]. Hyperleptinemia associated with the excessive accumulation of visceral fat has been reported to cause hypersensitivity to bacterial endotoxin via CD14 upregulation by STAT3 signaling in hepatocytes [31]. Recently, in the treatment of diabetes via lifestyle intervention, SGLT-2 inhibitors, or GLP-1, the association between visceral fat accumulation and MASLD pathophysiologies that are independent of type 2 diabetes has attracted attention [32,33].

Dietary habits contribute to the development of MASLD and MASH, not only in terms of quantity and caloric intake, but also in terms of specific nutrients. Fructose contributes to the development of MASH by increasing oxidative stress and tumor necrosis factor (TNF)-α [34,35]. Conversely, coffee has a protective effect on patients with MASLD through its antioxidant properties [36]. Thus, a family physician could prescribe dietary therapy to improve MASLD and MASH, such as calorie restriction, fructose restriction, dietary modification, and improvement of lifestyle-related diseases, including diabetes.

Alterations in the gut microbiota lead to increased intestinal permeability, increased fatty acid absorption, and increased circulating levels of molecules that contribute to the activation of inflammatory pathways and the release of proinflammatory cytokines, such as interleukin-6 and TNF-α [37]. Endoplasmic reticulum stress has been suggested to be involved in the pathogenesis of MASLD/MASH. Saturated fatty acids, hyperinsulinemia, inflammatory conditions, oxidative stress and ER stress are thought to be factors in the pathogenesis of MASLD/MASH. ER stress promotes fatty acid synthesis by activating SREBP-1c, suppresses VLDL production by inhibiting translation and degradation of apoB-100, and enhances fat accumulation in hepatocytes. Therefore, at least part of the cellular damage caused by ER stress may occur through oxidative stress-mediated mechanisms [38,39].

The single nucleotide polymorphism of the patatin-like phospholipase domain-containing 3 (*PNPLA3*) gene, a disease susceptibility gene polymorphism in MASLD/MASH and rs738409 (I148M). Racial differences also exist, with Mexican, Hispanic, and Japanese individuals reported to have a higher frequency of risk alleles than other ethnic groups [40,41].

In addition to *PNPLA3*, transmembrane 6 superfamily member [42], glucokinase regulatory protein, membrane-bound O-acyltransferase domain containing 7, hydroxysteroid 17-beta dehydrogenase 13, and other genes involved in glucose and lipid metabolic pathways have been implicated in MASLD susceptibility (Figure 2).

Further, in addition to acquired environmental factors, genetic backgrounds related to metabolism also affect the pathogenesis of MASLD/MASH, which may lead to the development of therapeutic strategies in the future.

## 4. Epidemiology

The prevalence of MASLD is increasing worldwide with the increase in the obese population and the number of patients with type 2 diabetes mellitus. In 2023, a meta-analysis of 92 studies (N = 9,361,716) found a global prevalence of MASLD of 30% [4], which is a clear increase from the previously reported 25% in 2016 [43]. The prevalence of MASLD has been reported to be 44.4%, 36.5, 33.8%, 33%, 31%, 30%, 28%, and 25% in Latin America, the Middle East and North Africa, South Asia, Southeast Asia, North America, East Asia, Asia Pacific, and Western Europe, respectively [4]. A Japanese meta-analysis has also predicted that the prevalence of MASLD will be more than 40% by 2040 [44].

The number of MASLD cases is expected to increase by 2030 according to the Markov model. Additionally, the prevalence of MASLD is expected to increase the most in China and the least in Japan. However, in Japan, liver mortality and progressive liver disease due to MASLD are expected to more than double due to the population aging [45]. As aforementioned, there are susceptibility genes for MASLD, especially the *PNPLA3* polymorphism, which varies by region [3]. Although the prevalence of MASLD is not defined by genes alone, those at genetic risk may also be associated with progression of MASLD, including the development of HCC after cirrhosis [43]. Although it is difficult to determine the prevalence in the general population, a prospective American study in 2021 identified MASLD in 38% and MASH in 14% of asymptomatic patients [46]. The study also revealed a more than two-fold increase in the prevalence of clinically significant fibrosis (stage 2 or higher fibrosis) [47]. Therefore, the incidence of liver failure, HCC, and other deaths associated with MASH-derived cirrhosis is expected to increase two- to three-fold by 2030 [46]. MASH-derived cirrhosis is already a major indication for liver transplantation in the United States in patients older than 65 years of age [47].

## 5. Noninvasive Tests (NITs) for Hepatic Fibrosis in Patients with MASLD

It is not practical to make a pathological diagnosis by liver biopsy in all patients; thus, it is important to select the group of patients with advanced fibrosis by using a simple and less invasive method. For this purpose, NITs are expected to be used. In particular, NITs using blood biochemical tests are useful for screening in primary care and include direct markers, which measure substances directly related to liver fibrosis metabolism, and indirect markers, which are calculated by combining routinely measured laboratory values [48].

Direct markers include type IV collagen 7s domain, ELF test, hyaluronic acid, Mac2 binding protein glucosylation isomer (M2BPGi), and type III procollagen peptide (PIIIP) [49,50,51,52,53]. In Europe and the United States, the ELF test, which combines hyaluronic acid, PIIIP, and the tissue inhibitor of matrix metalloproteinase 1, is considered to have high diagnostic performance [50]. However, there are currently no markers with high diagnostic performance that are recognized worldwide.

Indirect markers include fibrosis-4 (FIB-4) index, NAFLD fibrosis score (NFS), and platelets. The FIB-4 index can be calculated using the aspartate transaminase level, ALT level, platelet count, and patient age, and can be evaluated in general practice [54]. The FIB-4 index is strongly recommended by many guidelines as a simple and efficient tool to detect advanced fibrosis in patients with MASLD by combining items that are measured in daily practice [11,12,13,14]. The risk groups for liver fibrosis are low (<1.30), intermediate (1.30–2.66), and high (≥2.67). In a Japanese report, the negative predictive value of advanced fibrosis was 99% in the low-risk group (<1.30), which is useful for the primary screening of MASLD [55]. The aforementioned report also uses a cutoff value for the FIB-4 index because, in settings such as primary care, where the prevalence of advanced liver fibrosis is low, the emphasis is on exclusion by testing with a high negative predictive value.

The AASLD Practice Guidance 2023 similarly recommends referral to a specialized institution if the FIB-4 index is ≥1.3 but is more explicit about follow-up in the primary care setting for low-risk groups (<1.30). It recommends measuring the FIB-4 index every 1–2 years if diabetes or two or more metabolic factors are present, and every 2–3 years if not [11].

The NAFLD fibrosis score (NFS) uses six variables: age, presence of hyperglycemia, body mass index (BMI), platelet count, albumin, and the AST/ALT ratio [56]. Advanced fibrosis is ruled out with a score of <−1.45 [57]. Patients with a score of >−1.45 are suspected to have fibrosis progression, and a secondary examination is recommended. However, NFS may overestimate fibrosis in obese patients, leading to differences in diagnostic performance by race and region. Platelets are known to be the easiest direct marker for the detection of advanced fibrosis, because they decrease with the progression of liver fibrosis. The cut-off value of platelets by which to distinguish cases of stage 3 or higher liver fibrosis is reported to be 19.2 × 10^4^ /μL [58]. Cases of MASLD with platelets less than 20 × 10^4^ /μL should be examined secondarily.

## 6. Secondary Check and Detailed Examination by a Gastroenterologist/Hepatologist

The secondary check focuses on identifying high-risk patients with advanced liver fibrosis, including cirrhosis (Figure 3).

The AASLD Practice Guidance 2023 defines stage 2 and above as “at-risk NASH” with a high risk of liver disease-related events and death [11]. Identifying patients with at-risk NASH or advanced fibrosis can help identify cases that require close examination, such as liver biopsy. The development of algorithms for the assessment of at-risk NASH is essential to estimate the likelihood of progression from NASH to cirrhosis and HCC and to guide treatment. When the FIB-4 index is ≥1.3, the Japan Society of Gastroenterology guidelines recommend elastography and a liver biopsy [14]. Most international guidelines recommend vibration-controlled transient elastography (VCTE), an ultrasound elastography, as the next step after advanced fibrosis of MASLD has been ruled out by the FIB-4 index [11,12,13,14]. Transient elastography was developed as a modality to quantify liver fibrosis from the earliest stage and is frequently used worldwide for chronic liver diseases including MASLD. Approval to market for VCTE was received in China in 2008, Canada in 2009, Brazil in 2010, Japan in 2011, and the United States in 2013. As of October 2023, there are 260 VCTE units in Japan. In a meta-analysis of 37 studies and 5735 patients, the area under the receiver operating characteristic curves (AUROCs) of VCTE and the FIB-4 index in patients with advanced fibrosis of stage 3 or higher were 0.85 and 0.76, respectively [59]. Sequentially combining markers with lower cutoff values that exclude advanced fibrosis with markers with higher cutoff values that exclude cirrhosis can reduce the need for liver biopsy [60]. A two-step diagnosis combining FIB-4 index and VCTE has also been proposed [60]. A stepwise risk stratification model that combines FIB-4 (≥1.3) and LSM with VCTE (≥8 kPa) has been reported to save up to 87% of additional assessments [54].The time for one execution of VCTE is short and reproducible and follow-up can be performed as needed according to the patient’s clinical course. Simple non-invasive tests as well as histologically assessed fibrosis for predicting clinical outcomes in patients with MASLD could be considered as alternatives to liver biopsy in some cases [61]. The gastroenterologist/hepatopathologist should determine when to perform the first and second VCTE and to consider the presence/absence of indications for magnetic resonance elastography (MRE) and liver biopsy (Figure 4). MRE is currently considered the most accurate method by which to assess the status of liver fibrosis in patients with MASLD and has been used in drug trials. As of October 2023, there are 35 MRE units in Japan. In a meta-analysis of MASLD, the AUROC values for the diagnostic performance of MRE were reported to be 0.87 for F2 or higher, 0.90 for F3 or higher, and 0.91 for F4 [62]. Recently, the usefulness of the MRE combined with FIB-4 (MEFIB) index has been reported. The MEFIB index can narrow down at-risk NASH when the FIB-4 index is 1.6 or higher and the MRE is 3.3 or higher. This enables a highly accurate diagnosis with a positive predictive value of more than 95% [63]. Furthermore, not only does MRE have higher diagnostic performance than VCTE, but it can also measure a wider area of the liver than VCTE or a liver biopsy, which is a major advantage for screening for HCC. However, MRE lacks versatility owing to the high cost and low penetration rate.

Finally, liver biopsy is still considered the gold standard in many international guidelines for diagnosing MASH and is included in the latest EASL guidelines updated in 2021 [64] and AASLD guidelines updated in 2023 [11]. Because it is difficult to perform liver biopsy in all patients with NAFLD, we should use primary screening and elastography to narrow down the number of patients with advanced liver fibrosis, as shown in Figure 3. Liver biopsy should also be considered when discordance occurs in the evaluation by NIT, including elastography. Patients with fibrosis stage 0–1 should also be reevaluated periodically, and it is recommended that patients with at-risk NASH undergo annual fibrosis evaluation. Patients with cirrhosis require not only screening for varices and monitoring for signs of liver failure, but also semi-annual screening for hepatocellular carcinoma. Routine imaging studies are also required for cases with fibrosis development, while cases of NAFLD with low fibrosis do not require surveillance for HCC [11]. However, NAFLD in men, the elderly, and patients with diabetes mellitus, who are considered at risk for carcinogenesis, requires attention [65].

## 7. Collaboration between Family Doctors and Specialists for Liver Disease in Japan

Typical conditions that cause liver injury include viral hepatitis, alcoholic liver disease, drug-induced liver injury, autoimmune hepatitis (AIH), primary biliary cholangitis (PBC), and other metabolic diseases (e.g., thyroid dysfunction and thyroid cancer) [11,14]. Patients with steatotic liver disease detected by imaging should undergo examination for the cause of steatosis, with the important causes of steatotic liver disease, other than metabolic syndrome, being alcoholic steatotic liver disease and drug-induced liver injury [14]. Alcoholic steatotic liver disease is defined as alcohol consumption of 30 g/day or more in men and 20 g/day or more in women [11,14]. A history of over-the-counter medications and dietary supplements should be obtained as well as confirmation of prescription medications. Specifically, drug-induced liver injury, such as steatosis caused by tamoxifen use, should be excluded [11,14,66].

AIH and PBC could be assessed by testing for antinuclear antibodies, anti-mitochondrial antibodies, immunoglobulin (Ig)-G, and IgM. However, it should be noted that antinuclear antibodies may be positive or MASLD/MASH may be present even if the AIH score is met [67,68]. In Japan, a declaration was issued by the Japan Society of Hepatology in 2023 for the general public to see their family physician if they have an alanine transaminase (ALT) level of 30 or higher (Figure 5) [69]. In Kanagawa, the second largest prefecture in Japan by population, the Kanagawa Society of Internal Medicine developed and used a simplified information transfer sheet for liver disease treatment in cooperation with family physicians and hepatologists in 2014 (Figure 5).

## 8. Conclusions

The term NAFLD is being revised to MASLD owing to its reliance on exclusionary confounding and potentially stigmatizing language [1,2]; Japan has officially decided to use the term MASLD instead of NAFLD, while taking great care to maintain existing data on natural history, biomarkers, and clinical trials. In total, 98% of NAFLD patients have been shown to meet the new criteria for MASLD [70], and patients with the traditional definition (NAFLD) will be included in MASLD [1,2].

In Japan, the number of patients with MASLD is not expected to change significantly, but the rate of advanced fibrosis is expected to increase [44]. Primary care physicians must always be careful not to overlook MASLD cases with advanced fibrosis from a large number of eligible patients, mainly those with lifestyle-related diseases. The FIB-4 index, which does not require special blood tests or special medical equipment, has gained importance as the first method for primary care physicians to screen for cases of MASLD with advanced fibrosis. Patients with MASLD and without advanced liver fibrosis should also be followed up under the care of a regular family physician. Reassessment of the FIB-4 index is recommended every 2–3 years in patients without metabolic risk factors and every 1–2 years in patients with metabolic risk factors. Collaboration between family physicians and hepatologists is necessary at each stage of the disease process, from check-up and secondary screening to treatment and follow-up.

As a result of a thorough examination by a hepatologist, patients with MASLD who have not progressed to fibrosis are likely to be followed up by their family physician. Collaboration between family physicians and hepatologists is necessary at each stage of the disease process, from check-up and secondary screening to treatment and follow-up. Additionally, MASLD is frequently complicated by lifestyle-related diseases, such as obesity, diabetes, hypertension, and dyslipidemia, so it is necessary for hepatologists to collaborate with family physicians regarding associated complications.

In a knowledge survey in 40 countries, a large gap was found between primary care physicians and hepatologists in the knowledge of MASLD and its management [71]. The reasons for difficulties in collaboration between primary care physicians and hepatologists may include differences in goals, educational backgrounds, and cultures along with a lack of understanding of other professions. Primary care physicians generally rely on the internet as their primary source of knowledge [64]. Accurate information about MASLD needs to be provided to primary care physicians through online computer-based education modules and regular webinars on new updates. Furthermore, face-to-face meetings are also important for relationship building.

MASLD has become an important health problem from both medical and socioeconomic perspectives; however, it is a systemic disease, and its ideal management requires the involvement of many health care professionals, not just family physicians and hepatologists/gastroenterologists. MASLD has attracted attention as a systemic disease due to its pathogenic mechanism, poor outcomes, and crosstalk with other organs. If MASLD/MASH and the associated complications are not proactively screened, they may progress unnoticed. However, owing to the large number of potential patients, gastroenterologists and hepatologists cannot manage the patients alone, and collaboration between specialists in various fields, including family doctors, dentists, nutritionists, and pharmacists is required for treatment of NAFLD.

There are limitations in the evidence for MASLD provided in this study. There are differences in population and medical resources worldwide, and it is difficult to standardize the algorithm for MASLD management. In a previous report, regional differences in NIT assessing disease severity including VCTE were reported [72]. In the future, it is important to construct algorithms and a treatment strategy that can compensate for these environmental differences and that are tailored to the level of the medical facility. Furthermore, hepatologists need to collaborate with primary care physicians and other health care providers to clarify treatment goals and follow-up indicators in the treatment of MASLD.

## Figures and Tables

**Figure 1 life-13-02144-f001:**
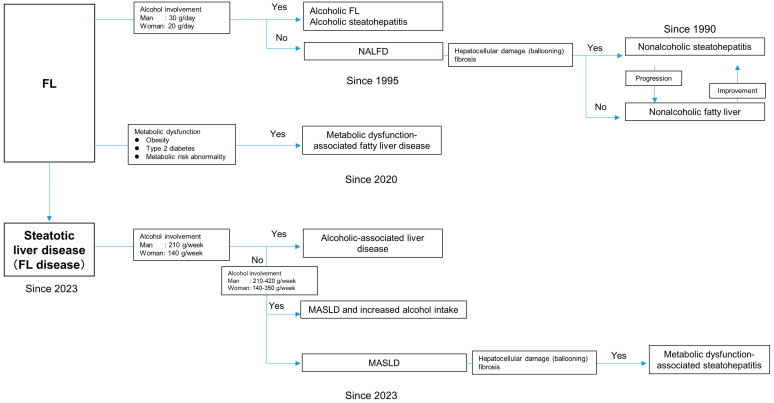
Changes in the concepts of NAFLD, MAFLD, and MASLD. NAFLD: nonalcoholic fatty liver disease, MAFLD: metabolic dysfunction-associated fatty liver disease, MASLD: metabolic dysfunction-associated steatotic liver disease; FL, fatty liver.

**Figure 2 life-13-02144-f002:**
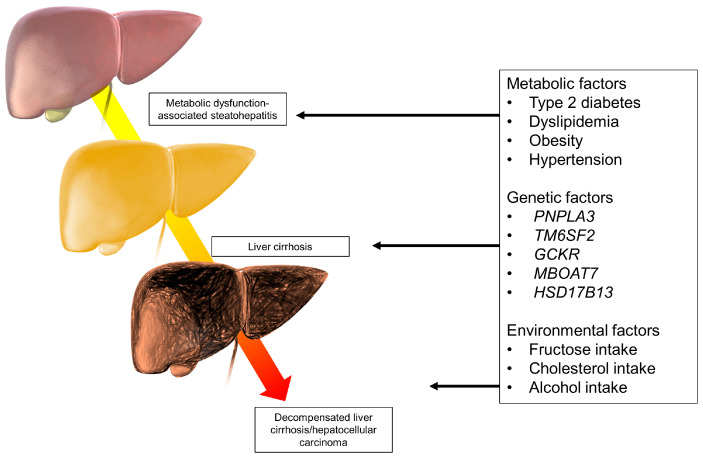
Metabolic, genetic, and lifestyle factors involved in the development and progression of MASLD. MASLD: metabolic dysfunction-associated steatotic liver disease.

**Figure 3 life-13-02144-f003:**
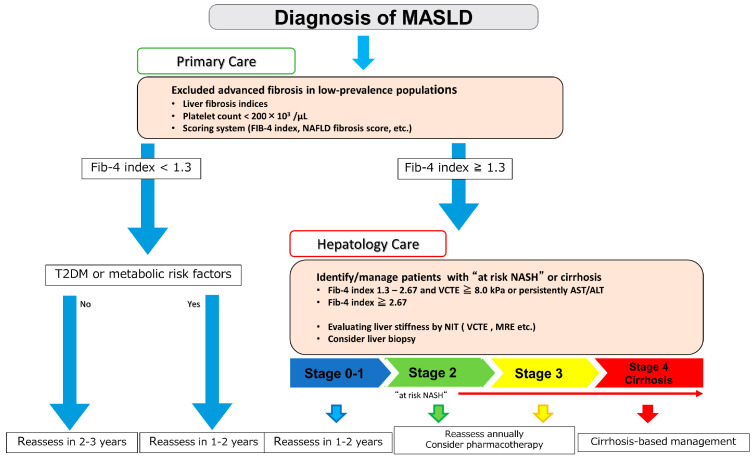
Two-step diagnostic algorithm used by primary care physicians and hepatologists to select cases of MASLD with advanced fibrosis. AST, aspartate transaminase; ALT, alanine transaminase; FIB-4, Fibrosis-4; MASLD: metabolic dysfunction-associated steatotic liver disease; MRE, magnetic resonance elastography; NASH, nonalcoholic steatohepatitis; T2DM, type 2 diabetes mellitus; VCTE, vibration-controlled transient elastography.

**Figure 4 life-13-02144-f004:**
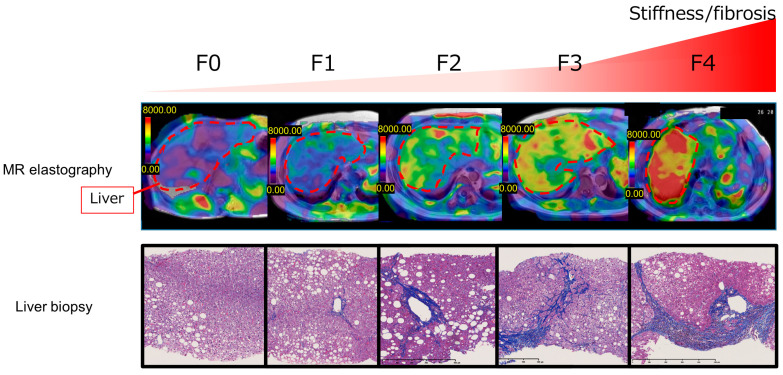
MR elastography and histology of patients with MASLD by each grade of fibrosis/hepatic stiffness. MASLD: metabolic dysfunction-associated steatotic liver disease; MR, magnetic resonance.

**Figure 5 life-13-02144-f005:**
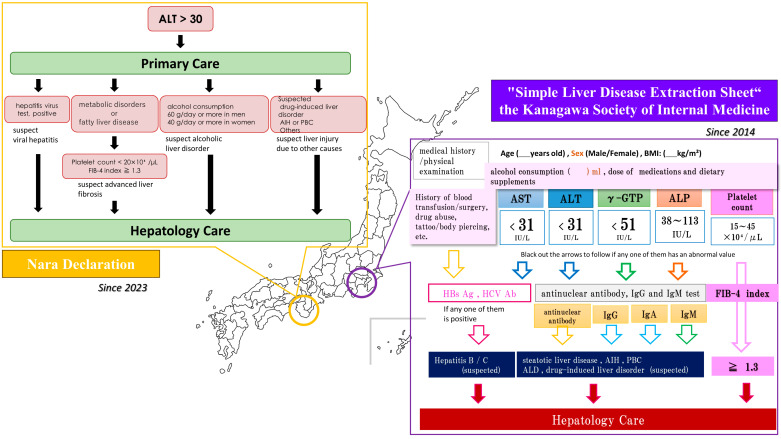
Specific examples of collaboration between primary care physicians and hepatologists in MASLD management in Japan. Shown here are the simple check sheet recommended by the Kanagawa Society of Internal Medicine since 2014 and the “Nara Declaration” published by the Japan Society of Hepatology in 2023. AST, aspartate transaminase; ALT, alanine transaminase; γ-GTP, gamma-glutamyl transpeptidase; ALP, alkaline phosphatase; Ig, immunoglobulin; HBs Ag, hepatitis B surface antigen; HCV, Ab, hepatitis C antibody; FIB-4, Fibrosis-4.

## Data Availability

Not applicable.

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
