# Peer review of "Frontiers of Collaboration between Primary Care and Specialists in the Management of Metabolic Dysfunction-Associated Steatotic Liver Disease: A Review"

_life, 2023, doi:10.3390/life13112144_

Round 1
Reviewer 1 Report
Comments and Suggestions for Authors
This review article about the collaboration between primary care and specialists in the management of MASLD. MASLD emphasizes metabolic dysfunction as the key association and steatohepatitis as the main concern rather than simple steatosis. MASLD is more specific to progressive disease but very newly introduced. A comprehensive overview of the latest findings and guidelines on MASLD is provided in the article, including its definition, pathogenesis, diagnosis, and treatment. The role, accuracy, and clinical use of non-invasive tests for evaluating liver fibrosis, which is a key factor for prognosis and management of MASLD, are discussed. A strategy for screening advanced fibrosis cases in primary care in Japan is introduced, and ways to improve the collaboration between primary care physicians and hepatologists are suggested. Relevant and recent studies from various sources and countries are cited to support the arguments and recommendations made.
However, a few concerns need to be addressed concisely:
- a clear, concise summary or conclusion highlighting the main points and implications for “practice” is not provided at the end of the article
- the potential challenges or barriers for implementing the proposed strategy or improving the collaboration between primary care and specialists are not addressed
- different non-invasive tests or guidelines for MASLD are also not concisely compared or contrasted in terms of their pros, cons, or applicability in different settings or populations
- the limitations or gaps in the current evidence or research on MASLD are not discussed, and directions for future studies are not suggested
Comments on the Quality of English Language
using the future tense "will discuss" in the conclusion is not correct?
Author Response
Dear Prof. Lluís Ribas de Pouplana, Editor, Life, Dr. Kosmalski Marcin and Dr. Łukasz Mokros, Guest Editor, Dr. Danika Yuan, Assistant Editor, Life
October 27th, 2023
We would like to express our sincere gratitude to both you and the Reviewers for taking the time to review our manuscript titled “Frontiers of collaboration between primary care and specialists in the management of metabolic dysfunction-associated steatotic liver disease: A review” submitted to Life (Basel). The reviewers’ comments and Editor’s comments were very encouraging and helpful. We have now amended our manuscript according to your suggestions and the Reviewer’s comments. Please find below the changes we have made to the manuscript and our answers to the reviewers’ comments in a point-by-point manner.
Response to Reviewer 1
We thank the Reviewer for their insightful suggestions that have helped us improve our manuscript significantly. We have now addressed your comments in the revised manuscript. We hope you will find our responses and changes to the manuscript satisfactory.
Comment 1 by Reviewer 1
- a clear, concise summary or conclusion highlighting the main points and implications for “practice” is not provided at the end of the article
Response to comment 1 by Reviewer 1
Thank you for your very valuable comments. We have revised the last sentence as follows: "Hepatologists need to collaborate with primary care physicians and other health care providers to clarify treatment goals and follow-up indicators in the treatment of MASLD.
Comment 2 by Reviewer 1
- the potential challenges or barriers for implementing the proposed strategy or improving the collaboration between primary care and specialists are not addressed
Response to comment 2 by Reviewer 1
Thank you for the comments. Reasons for difficulties in collaboration between primary care physicians and hepatologists may include differences in goals, educational backgrounds and cultures, and lack of understanding of other professions. We have added this statement to the Conclusion section (Page 10, Lines 374-376).
Comment 3 by Reviewer 1
- different non-invasive tests or guidelines for MASLD are also not concisely compared or contrasted in terms of their pros, cons, or applicability in different settings or populations
Response to comment 3 by Reviewer 1
Thank you for your excellent comment. As you pointed out, there are differences among environments and populations. In particular, MRE may be of limited use due to their low prevalence. In a previous report, it was reported that there are regional differences in NIT assessing disease severity including VCTE (JHEP Rep. 2021 Nov 22;4(1):100411). Therefore, it is necessary to take measures according to each region and environment. In addition, collaboration between primary care physicians, hepatologists, and other health care providers is necessary in each region. We have included this information in Page 7, Lines 291-292 and Page 10, Lines 391-398.
Comment 4 by Reviewer 1
- the limitations or gaps in the current evidence or research on MASLD are not discussed, and directions for future studies are not suggested
Response to comment 4 by Reviewer 1
Thank you for your comment. As you pointed out, there are limitations in the evidence for MASLD. There are differences in population and medical resources worldwide, and it is difficult to standardize the algorithm for MASLD treatment. In the future, it is necessary to construct an algorithm that can fill in these environmental differences according to the level of medical facilities, and this algorithm should be the basis for the development of new drugs and the establishment of screening for carcinogenesis. We have added this statement to the Conclusion section (Page 10, Lines 391-398).

Reviewer 2 Report
Comments and Suggestions for Authors
1. In part 3, the authors discussed the role of IR, diet, intestinal microbiota, and genetics in the pathogenesis of MASLD/MASH. Please discuss more, for example, the visceral adipose tissue and ER stress.
2. The title of part 5 and part 6 should be more summarized.
Comments on the Quality of English Language1. Language needs to be improved throughout the manuscript. Some grammatical errors and typos are noted. For example, in line 22, “livre”
Author Response
Dear Prof. Lluís Ribas de Pouplana, Editor, Life, Dr. Kosmalski Marcin and Dr. Łukasz Mokros, Guest Editor, Dr. Danika Yuan, Assistant Editor, Life
October 27th, 2023
We would like to express our sincere gratitude to both you and the Reviewers for taking the time to review our manuscript titled “Frontiers of collaboration between primary care and specialists in the management of metabolic dysfunction-associated steatotic liver disease: A review” submitted to Life (Basel). The reviewers’ comments and Editor’s comments were very encouraging and helpful. We have now amended our manuscript according to your suggestions and the Reviewer’s comments. Please find below the changes we have made to the manuscript and our answers to the reviewers’ comments in a point-by-point manner.
Response to Reviewer 2
We thank the Reviewer for their insightful suggestions that have helped us improve our manuscript significantly. We have now addressed your comments in the revised manuscript. We hope you will find our responses and changes to the manuscript satisfactory.
comment 1 by Reviewer 2.
In part 3, the authors discussed the role of IR, diet, intestinal microbiota, and genetics in the pathogenesis of MASLD/MASH. Please discuss more, for example, the visceral adipose tissue and ER stress.
Response to comment 1 by Reviewer 2.
Thank you for your kind comment. I added more discussion about visceral adipose tissue and ER stress in Part 3.
comment 2.
The title of part 5 and part 6 should be more summarized.
Response to comment 2 by Reviewer 2.
Thank you for your kind comment. Chapters have been changed and renamed to reflect the revision: 1. Introduction, 2. Concepts of NAFLD/NASH, MAFLD, and MASLD, 3. Pathogenesis of MASLD/MASH, 4. Epidemiology, 5. Noninvasive tests (NITs) for hepatic fibrosis in patients with MASLD, 6. Secondary check and detailed examination by a gastroenterologist/hepatologist, 7. The collaboration system between family doctors and specialists for liver disease in Japan. 8. Conclusions.
comment 3.
Language needs to be improved throughout the manuscript. Some grammatical errors and typos are noted. For example, in line 22, “livre”
Response to comment 3 by Reviewer 2.
Thank you very much for your kind comments. The text has been carefully revised and proofread again in English by Editage.

Reviewer 3 Report
Comments and Suggestions for Authors
A few comments for the authors:
1. This is a nice review paper for NAFLD. Although MASLD has now been adopted to replace NAFLD, I think a word of caution may be necessary about whether MASLD and NAFLD are the same disease with the same long term outcomes given the increase in the amount of alcohol allowed in the MASLD definition- and as the authors themselves point out the level of alcohol allowed in the MASLD definition may itself cause liver damage. However, with the caveat that if these definitions are proven to be similar, these are the things to consider etc can be acceptable.
2. I think the paper can be a little better organized in that you introduce the risk stratification used and then talk about the tests etc- it may be better to talk about the NITS first and then introduce the risk stratification suggested in Japan and why Japan needs different risk stratification guidelines than what have already been published-
3. Also, is NAFLD used in Japan or MAFLD so that MASLD would not be part of the nomenclature?
4. Parts of the summary seem to be better suited for the introduction rather than the summary especially when you read line 322- please consider redoing the summary and perhaps adding some of the information in the summary to the introduction and summarizing the highlights of your paper- what do you want people to know and or do.
5. Please correct the spelling of "liver" line 22 and remove the edits left in the manuscript.
Author Response
Dear Prof. Lluís Ribas de Pouplana, Editor, Life, Dr. Kosmalski Marcin and Dr. Łukasz Mokros, Guest Editor, Dr. Danika Yuan, Assistant Editor, Life
October 28th, 2023
We would like to express our sincere gratitude to both you and the Reviewers for taking the time to review our manuscript titled “Frontiers of collaboration between primary care and specialists in the management of metabolic dysfunction-associated steatotic liver disease: A review” submitted to Life (Basel). The reviewers’ comments and Editor’s comments were very encouraging and helpful. We have now amended our manuscript according to your suggestions and the Reviewer’s comments. Please find below the changes we have made to the manuscript and our answers to the reviewers’ comments in a point-by-point manner.
Response to Reviewer 3
We thank the Reviewer for their insightful suggestions that have helped us improve our manuscript significantly. We have now addressed your comments in the revised manuscript. We hope you will find our responses and changes to the manuscript satisfactory.
comment 1 by Reviewer 3.
This is a nice review paper for NAFLD. Although MASLD has now been adopted to replace NAFLD, I think a word of caution may be necessary about whether MASLD and NAFLD are the same disease with the same long term outcomes given the increase in the amount of alcohol allowed in the MASLD definition- and as the authors themselves point out the level of alcohol allowed in the MASLD definition may itself cause liver damage. However, with the caveat that if these definitions are proven to be similar, these are the things to consider etc can be acceptable.
Response to comment 1 by Reviewer 3.
Thank you very much for your kind comments. In accordance with the reviewer's comments, we have inserted the following text in the concluding paragraph of the text.
The term NAFLD is being renamed MASLD due to its reliance on exclusionary confounding and potentially stigmatising language [1, 2], and Japan has officially decided to change its name to MASLD. In changing the name from NAFLD to MASLD, great care was taken to maintain existing data on natural history, biomarkers, and clinical trials. 98% of NAFLD patients have been shown to meet the new criteria for MASLD [65], and patients with the traditional definition (NAFLD) will be included in MASLD. MASLD [1,2].
Comment 2.
I think the paper can be a little better organized in that you introduce the risk stratification used and then talk about the tests etc- it may be better to talk about the NITS first and then introduce the risk stratification suggested in Japan and why Japan needs different risk stratification guidelines than what have already been published-
Response to comment 2 by Reviewer 3.
Thank you for your kind comment. Chapters have been changed and renamed to reflect the revision: 1. Introduction, 2. Concepts of NAFLD/NASH, MAFLD, and MASLD, 3. Pathogenesis of MASLD/MASH, 4. Epidemiology, 5. Noninvasive tests (NITs) for hepatic fibrosis in patients with MASLD, 6. Secondary check and detailed examination by a gastroenterologist/hepatologist, 7. The collaboration system between family doctors and specialists for liver disease in Japan. 8. Conclusions.
Comment 3.
Also, is NAFLD used in Japan or MAFLD so that MASLD would not be part of the nomenclature?
Response to comment 3 by Reviewer 3.
Thank you for your kind comment. Since MAFLD was proposed in Japan in 2020, there has been an increase in the number of research reports related to MAFLD. However, MAFLD has been known only to a limited number of researchers. In October 2023, the Japanese Society of Gastroenterology and Hepatology and the Japanese Society of Hepatology declared to officially adopt MASLD proposed by AASLD and EASL. It is believed that MASLD will be the official for practice and research in Japan.
Comment 4.
Parts of the summary seem to be better suited for the introduction rather than the summary especially when you read line 322- please consider redoing the summary and perhaps adding some of the information in the summary to the introduction and summarizing the highlights of your paper- what do you want people to know and or do.
Response to comment 4 by Reviewer 3.
Thank you for your kind comment. In accordance with the reviewer, the Abstract was rewritten to reflect the content of the paper as a whole.
Comment 5.
Please correct the spelling of "liver" line 22 and remove the edits left in the manuscript.
Response to comment 5 by Reviewer 3.
Thank you very much for your kind comments. The text has been carefully revised and proofread again in English by Editage.
